# A General Algorithm for the Split Common Fixed Point Problem with Its Applications to Signal Processing

**Wachirapong Jirakitpuwapat** [1,†] ID**, Poom Kumam** [2,3,*,†] ID**, Yeol Je Cho** [2,4,5,†]
**and Kanokwan Sitthithakerngkiet** [6,†] ID

[1]  KMUTT-Fixed Point Research Laboratory, Room SCL 802 Fixed Point Laboratory, Science Laboratory
    Building, Department of Mathematics, Faculty of Science, King Mongkut's University of Technology
    Thonburi (KMUTT), 126 Pracha Uthit Rd., Bang Mod, Thung Khru, Bangkok 10140, Thailand;
    wachirapong.jira@hotmail.com
[2]  KMUTT-Fixed Point Theory and Applications Research Group, Theoretical and Computational Science
    Center (TaCS), Science Laboratory Building, Faculty of Science, King Mongkut's University of Technology
    Thonburi (KMUTT), 126 Pracha-Uthit Road, Bang Mod, Thrung Khru, Bangkok 10140, Thailand;
    yjchomath@gmail.com
[3]  Department of Medical Research, China Medical University Hospital, China Medical University,
    Taichung 40402, Taiwan
[4]  Department of Mathematics Education, Gyeongsang National University, Jinju 52828, Korea
[5]  School of Mathematical Sciences, University of Electronic Science and Technology of China,
    Chengdu 611731, China
[6]  Department of Mathematics, Faculty of Applied Science, King Mongkut's University of Technology North
    Bangkok (KMUTNB), Wongsawang, Bangsue, Bangkok 10800, Thailand; kanokwan.s@sci.kmutnb.ac.th
*   Correspondence: poom.kum@kmutt.ac.th; Tel.: +66-(0)2470-8994
†   These authors contributed equally to this work.

**Abstract:** In 2014, Cui and Wang constructed an algorithm for demicontractive operators and proved some weak convergence theorems of their proposed algorithm to show the existence of solutions for the split common fixed point problem without using the operator norm. By Cui and Wang's motivation, in 2015, Boikanyo constructed also a new algorithm for demicontractive operators and obtained some strong convergence theorems for this problem without using the operator norm. In this paper, we consider a viscosity iterative algorithm in Boikanyo's algorithm to approximate to a solution of this problem and prove some strong convergence theorems of our proposed algorithm to a solution of this problem. Finally, we apply our main results to some applications, signal processing and others and compare our algorithm with five algorithms such as Cui and Wang's algorithm, Boikanyo's algorithm, forward-backward splitting algorithm and the fast iterative shrinkage-thresholding algorithm (FISTA).

**Keywords:** split common fixed point problem; demicontractive operator; Cui and Wang's algorithm; Boikanyo's algorithm; strong convergence

**MSC:** 47J25; 47J20; 49N45; 65J15

## 1. Introduction

Assume that $C$ and $Q$ are nonempty closed convex subsets of Hilbert spaces $H_1$ and $H_2$, respectively. Assume that $A : H_1 \rightarrow H_2$ is a bounded linear operator with the adjoint $A^*$.

In 1994, the split feasibility problem was proposed by Censor and Elfving [1] as follows:

$$\text{Find a point } x^* \in H_1 \text{ such that } x^* \in C \text{ and } Ax^* \in Q. \tag{1}$$

It is interesting to note that, when taking $C = H_1$ and $Q = \{b\}$, the split feasibility problem reduces to the linear inverse problem:

$$\text{Find a point } x^* \in H_1 \text{ such that } Ax^* = b. \tag{2}$$

The most popular ways for solving the linear inverse problem is to reformulate it as a least squares problem. Similarly, the split feasibility problem was solved by equivalently reformulating it as the convex optimization problem:

$$\min_{x \in C} \frac{1}{2} \| Ax - P_Q(Ax) \|^2, \tag{3}$$

where $P_Q(\cdot)$ is the projection operator on set $Q$ defined by

$$P_Q(v) = \arg\min_{z \in Q} \| z - v \|.$$

In 2002, based on the reformulation (3), the so-called CQ algorithm was presented by Byrne. He solved this problem by using the algorithm: For an arbitrary $x_1 \in H$,

$$x_{n+1} = A^{-1} P_Q(P_{A(C)}(Ax_n)), \quad \forall n \in \mathbb{N}, \tag{4}$$

which converges to a solution of the convex optimization problem. Since the algorithm (4) requires the inverse matrix of $A$, it is disadvantage to calculate this algorithm. We note that $x^* \in H$ solves the problem (2) is equivalent to the fixed point problem, that is, $x^*$ is a fixed point of $T$, where $T := P_C(I - \rho A^*(I - P_Q)A)$ for any $\rho > 0$.

In 2002, Byrne [2] constructed the following algorithm (5), which does not compute the inverse matrix of $A$: For any $x_0$, $\{x_n\}$ is generated by

$$x_{n+1} = P_C(I - \rho A^*(I - P_Q)A)x_n, \quad \forall n \in \mathbb{N}, \tag{5}$$

where $\rho \in (0, \frac{2}{L})$ and $L$ is the largest eigenvalue of $A^*A$.

Recently, the split feasibility problem has been apllied to approximation theory, signal processing, image recovery, control theory, biomedical engineering, geophysics and communications by many authors. Refer to the papers [3–9].

Especially, the split common fixed point problem is as follows:

$$\text{Find a point } x^* \in H \text{ such that } x^* \in \text{Fix}(U) \text{ and } Ax^* \in \text{Fix}(T), \tag{6}$$

where $U : H \to H$ and $T : K \to K$ are operators, $\text{Fix}(U)$ and $\text{Fix}(T)$ denote the fixed point sets of $U$ and $T$, respectively. In 2009, this problem was proposed by Censor and Segal [10] and they constructed the following algorithm for solving the problem: For any $x_0 \in H$, $\{x_n\}$ is generated by

$$x_{n+1} = U(I - \rho A^*(I - T)A)x_n, \quad \forall n \in \mathbb{N}. \tag{7}$$

This algorithm can be extended to many cases as follows:

1.　Quasi-nonexpansive operators by Moudafi [11];
2.　Finitely many directed operators by Wang and Xu [12];

3. Demicontractive operators by Moudafi [13]. In the case when $U$ and $T$ are directed operators, the step size $\rho$ satisfies $0 < \rho < \frac{2}{\|A\|^2}$ and $\{x_n\}$ generated by the algorithm (7) converges weakly to a solution of the problem (6) when a solution exists.

The algorithm (7) needs to compute $\|A\|$, which is not easily computed. In 2014, Cui and Wang [14] proposed the following Algorithm 1 without using the operator norm: For an initial $x_0 \in H$,

$$x_{n+1} \;=\; U_\lambda(x_n - \rho_n A^*(I - T)Ax_n), \quad \forall n \geq 0, \tag{8}$$

where

$$\rho_n = \begin{cases} \frac{(1-\tau)\|(I-T)Ax_n\|^2}{2\|A^*(I-T)Ax_n\|^2}, & Ax_n \neq T(x_n), \\ 0 & \text{otherwise}, \end{cases}$$

where $U$ and $T$ are demicontractive operators with constants $0 \leq \kappa < 1$ and $0 \leq \tau < 1$ such that $I - U$ and $I - T$ are demiclosed at zero, respectively, denote $U_\lambda := (1 - \lambda)I + \lambda U$ for any $\lambda \in (0, 1 - \kappa)$ and $A$ is a bounded linear operator, and they proved that the algorithm (8) converges weakly to a solution of the problem (6) when a solution exists.

---

**Algorithm 1:** Cui and Wang's algorithm

**Input:** Set $\lambda \in (0, 1 - \kappa)$, where $0 \leq \kappa < 1$. Choose $x_0 \in H$.

1 **for** $n = 1, 2, \cdots$ **do**
2 　　Update $x_{n+1}$ via (8),
3 **end for**

---

In 2015, Boikanyo [15] extended Cui and Wang's results and proposed the following Algorithm 2 for demicontrative operators $U$ and $T$ with $U_\lambda := (1 - \lambda)I + \lambda U$ for any $\lambda \in (0, 1 - \kappa)$, which converges strongly to a solution of the problem (6) when a solution exists: For any $u \in H$,

$$x_{n+1} \;=\; \alpha_n u + (1 - \alpha_n)U_\lambda(x_n - \rho_n A^*(I - T)Ax_n), \quad \forall n \geq 0, \tag{9}$$

where

$$\rho_n = \begin{cases} \frac{(1-\tau)\|(I-T)Ax_n\|^2}{2\|A^*(I-T)Ax_n\|^2}, & Ax_n \neq T(x_n), \\ 0 & \text{otherwise}, \end{cases}$$

and $\{\alpha_n\}$ is a sequence in $[0, 1)$ such that

$$\lim_{n\to\infty} \alpha_n = 0, \quad \sum_{n=0}^{\infty} \alpha_n = \infty.$$

---

**Algorithm 2:** Boikanyo's algorithm

**Input:** Set $\lambda \in (0, 1 - \kappa)$ where $0 \leq \kappa < 1$, and $\alpha_n \in [0, 1)$ such that $\lim_{n\to\infty} \alpha_n = 0$ and $\sum_{n=0}^{\infty} \alpha_n = \infty$. Choose $u, x_0 \in H$.

1 **for** $n = 1, 2, \cdots$ **do**
2 　　Update $x_{n+1}$ via (9).
3 **end for**

---

In 2016, Huimin et al. [16] proposed the following Algorithm 3 for demicontrative operators $U, T$ with $U_\lambda := (1 - \lambda)I + \lambda U$ for any $\lambda \in (0, 1 - \kappa)$, where $\lambda \in (0, 1 - \kappa)$, and $f$ is a contraction operator on $\text{Fix}(U)$ which converges strongly to a solution of the problem (6) when a solution exists:

$$x_{n+1} = \alpha_n f(x_n) + (1 - \alpha_n)U_\lambda(x_n - \rho_n A^*(I - T)Ax_n), \quad \forall n \geq 0, \tag{10}$$

where

$$\rho_n = \begin{cases} \frac{(1-\tau)\|(I-T)Ax_n\|^2}{2\|A^*(I-T)Ax_n\|^2}, & Ax_n \neq T(x_n), \\ 0 & \text{otherwise,} \end{cases}$$

and $\{\alpha_n\}$ is a sequence in $[0, 1)$ such that

$$\lim_{n \to \infty} \alpha_n = 0, \quad \sum_{n=0}^{\infty} \alpha_n = \infty.$$

---

**Algorithm 3:** Algorithm of Huimin et al. [16]

---

**Input:** Set $\lambda \in (0, 1 - \kappa)$, where $\lambda \in (0, 1 - \kappa)$, and $\alpha_n \in [0, 1)$ such that $\lim_{n \to \infty} \alpha_n = 0$ and

$\sum_{n=0}^{\infty} \alpha_n = \infty$. Choose $u, x_0 \in H$.

1  **for** $n = 1, 2, \cdots$ **do**
2      Update $x_{n+1}$ via (10).
3  **end for**

---

In this paper, motivated by Boikanyo's algorithm [15] and the algorithm of Huimin et al. [16], we will propose the following Algorithm 4 for demicontrative operators $U$ and $T$ with $U_{\lambda_n} := (1 - \lambda_n)I + \lambda_n U$ for any $\lambda_n \in (0, 1 - \kappa)$:

$$\begin{cases} y_n = \alpha_n f(x_n) + (1 - \alpha_n)U_{\lambda_n}(x_n - \rho_n A^*(I - T)Ax_n), \\ x_{n+1} = (1 - \beta_n)y_n + \beta_n f(y_n), \quad \forall n \geq 0, \end{cases} \tag{11}$$

where

$$\rho_n = \begin{cases} \frac{(1-\tau)\|(I-T)Ax_n\|^2}{2\|A^*(I-T)Ax_n\|^2}, & Ax_n \neq T(x_n), \\ 0 & \text{otherwise,} \end{cases}$$

$U$ and $T$ are demicontrative operators such that $I - U$ and $I - T$ are demiclosed at zero, $f$ is a contraction operator on $\text{Fix}(U)$ and the sequences $\{\alpha_n\}$, $\{\beta_n\}$ in $[0, 1)$ are such that

$$\lim_{n \to \infty} \alpha_n = 0, \quad \sum_{n=0}^{\infty} \alpha_n = \infty, \quad \sum_{n=0}^{\infty} \beta_n < \infty$$

and we prove that our algorithm $\{x_n\}$ generated by (11) converges strongly to a solution of the problem (6) when a solution exists. However, $\{x_n\}$ and $\{y_n\}$ converge to the same point because from the condition $0 \leq \beta_n < 1$ and $\sum_{n=1}^{\infty} \beta_n < \infty$.

---

**Algorithm 4:** Our algorithm

---

**Input:** Set $\lambda_n \in (0, 1 - \kappa)$, where $\lambda \in (0, 1 - \kappa), \alpha_n, \beta_n \in [0, 1)$ such that $\lim\limits_{n\to\infty} \alpha_n = 0$,

$\sum\limits_{n=0}^{\infty} \alpha_n = \infty$ and $\sum\limits_{n=0}^{\infty} \beta_n < \infty$. Choose $x_0 \in H$;

1 for each $n = 1, 2, \cdots$ **do;**

2 Update $y_n$ and $x_{n+1}$ via (11), respectively.

3 **end for**

---

**Remark 1.** *In fact, our algorithm was changed from the algorithm of Huimin et al. including the point u in Boikano's algorithm to the viscosity term and linear convex combination. The algorithm of Huimin et al. is a special case of our algorithm when $\beta_n = 0$ and $\{\lambda_n\}$ is a constant sequence. The algorithm of Huimin et al. and our algorithm are different because they were generated the distinct terms $x_n$. However, they converge strongly to a same solution of the split common fixed point problem.*

*For example, let*

$$y = \begin{bmatrix} 1.5 & 7 \end{bmatrix}^\dagger, \quad \epsilon = \begin{bmatrix} 0.5 & 1 \end{bmatrix}^\dagger, \quad A = \begin{bmatrix} 1 & 0 & 0 \\ 1 & 2 & 3 \end{bmatrix},$$

$$\alpha_n = \frac{0.1}{n}, \quad \beta_n = \frac{1}{n^2}, \quad \lambda_n = \frac{1}{2},$$

$$f(x) = \frac{x - \begin{bmatrix} 2 & 1 & 0 \end{bmatrix}^\dagger}{4} + \begin{bmatrix} 2 & 1 & 0 \end{bmatrix}^\dagger, \quad t = 10,$$

*where $^\dagger$ is transpose. If $x_{Our,100} = \begin{bmatrix} 1.5024 & 1.4672 & 0.8540 \end{bmatrix}^\dagger$ is generated by our algorithm and $x_{H,100} = \begin{bmatrix} 1.5034 & 1.0701 & 1.1177 \end{bmatrix}^\dagger$ is generated by the algorithm of Huimin et al., then two algorithms, the algorithm of Huimin et al. (10) and our algorithm (11) converge strongly to a same solution of the problem (15).*

## 2. Preliminaries

Let $H$ be a real Hilbert space. Let $x_n \rightharpoonup x$ denote that $\{x_n\}$ converges weakly to $x$ and $x_n \to x$ denote that $\{x_n\}$ converges strongly to $x$.

The following inequality holds:

$$\|x + y\|^2 \leq \|x\|^2 + 2\langle y, x + y \rangle, \quad \forall x, y \in H.$$

**Definition 1.** *Let $T : H \to H$ be an operator such that $\text{Fix}(T) \neq \emptyset$. Then $T$ is said to be:*

1. *Nonexpansive if*
$$\|Tx - Ty\| \leq \|x - y\|, \quad \forall x, y \in H;$$

2. *Contractive if there exists $k \in [0, 1)$ such that*
$$\|Tx - Ty\| \leq k\|x - y\|, \quad \forall x, y \in H;$$

3. *Quasi-nonexpansive if*
$$\|Tx - x^*\| \leq \|x - x^*\|, \quad \forall x, y \in H, x^* \in \text{Fix}(T);$$

4. *Directed if*
$$\|x^* - Tx\|^2 + \|x - Tx\|^2 - \|x - x^*\|^2 \leq 0, \quad \forall x, y \in H, x^* \in \text{Fix}(T);$$

5.  *τ-demicontractive with τ ∈ [0, 1) if*

$$\|Tx - x^*\|^2 \leq \|x - x^*\|^2 + \tau\|x - Tx\|^2, \quad \forall x, y \in H, \; x^* \in \text{Fix}(T).$$

**Remark 2.** *Easily, we obtain the following conclusions:*

1.  *Every contraction operator is nonexpansive;*
2.  *Every nonexpansive operator is quasi-nonexpansive;*
3.  *Every quasi-nonexpansive operator is 0-demicontractive operator;*
4.  *Every direct operator is −1-demicontractive operator.*

**Definition 2.** *Assume that $T : H \to H$ is an operator. Then $I - T$ is demiclosed at zero if, for any $\{x_n\}$ in H, $x_n \rightharpoonup x^*$ and $(I - T)x_n \to 0$ imply $Tx^* = x^*$.*

**Remark 3.** *Every nonexpansive operator is demiclosed at zero [17].*

**Definition 3.** *Assume that C is a nonempty closed convex subset of H. The metric projection $P_C$ from H onto C is defined as follows: For all $x \in H$,*

$$\|x - P_C x\| = \inf\{\|x - y\| : y \in C\}.$$

*Note that the metric projection $P_C$ is nonexpansive [17].*

**Lemma 1** ([18])**.** *Assume that C is a nonempty closed convex subset of H and $P_C$ is a nonexpansive operator from H onto C. For any $x \in H$, it satisfies the inequality:*

$$\langle P_C x - x, P_C x - y \rangle \leq 0, \quad \forall y \in C.$$

**Lemma 2** ([19])**.** *Assume that $\{\alpha_n\}$ is a sequence of nonnegative numbers such that*

$$\alpha_{n+1} \leq (1 - \beta_n)\alpha_n + \gamma_n, \quad \forall n \geq 0,$$

*where $\beta_n \in (0, 1)$ and $\gamma_n \in \mathbb{R}$ such that*

1.  $\sum_{n=1}^{\infty} \beta_n = \infty$;
2.  $\limsup_{n\to\infty} \frac{\gamma_n}{\beta_n} \leq 0$ *or* $\sum_{n=1}^{\infty} |\gamma_n| < \infty$.

*Then* $\lim_{n\to\infty} \alpha_n = 0.$

**Lemma 3** ([20])**.** *Assume that $A : H \to H$ is a τ-demicontractive operator with $\tau < 1$. Define $U_\lambda := (1 - \lambda)I + \lambda U$ for any $\lambda \in (0, 1 - \tau)$. Then, for any $x \in H$ and $x^* \in \text{Fix}(U)$,*

$$\|U_\lambda x - x^*\|^2 \leq \|x - x^*\|^2 - \lambda(1 - \tau - \lambda)\|x - Ux\|^2.$$

**Lemma 4** ([14])**.** *Assume that $A : H \to H$ is a bounded linear operator. Assume that $T : H \to H$ is a τ-demicontractive operator. If $A^{-1}(\text{Fix}(T)) \neq \emptyset$, then*

1.  $(I - T)Ax = 0$ *if and only if* $A^*(I - T)Ax = 0$ *for all $x \in H$;*
2.  *In particular, for all $x^* \in A^{-1}(\text{Fix}(T))$,*

$$\|x - \rho A^*(I - T)Ax - x^*\|^2 \leq \|x - x^*\|^2 - \frac{(1 - \tau)^2\|(I - T)Ax\|^4}{4\|A^*(I - T)Ax\|^2},$$

*where $x \in H$, $Ax \neq T(Ax)$ and*

$$\rho = \frac{(1-\tau)\|(I-T)Ax\|^2}{2\|A^*(I-T)Ax\|^2}.$$

## 3. Main Results

**Theorem 1.** *Assume that $H_1$ and $H_2$ are real Hilbert spaces. Assume that $U : H_1 \to H_1$ and $T : H_2 \to H_2$ are a $\kappa$-demicontractive operator and a $\tau$-demicontractive operator with constants $0 \leq \kappa < 1$ and $0 \leq \tau < 1$, respectively such that $I - U$ and $I - T$ are demiclosed at zero, respectively. Assume that $A : H_1 \to H_2$ is a bounded linear operator with the adjoint $A^*$ of $A$. Assume that $f$ is a contraction operator with constant $\eta$. Assume that $S$ is a set of all solution of the problem (6) such that $S \neq \emptyset$. If $\lim\limits_{n\to\infty} \alpha_n = 0$, $\sum\limits_{n=0}^{\infty} \alpha_n = \infty$ and $\sum\limits_{n=0}^{\infty} \beta_n < \infty$, then the sequence $\{x_n\}$ generated by algorithm (11) converges strongly to a point $x^* \in S$, which is a solution $x^* = P_S f(x^*)$ of the following variational inequality:*

$$\langle x^* - f(x^*), x^* - z \rangle \leq 0, \quad \forall z \in S. \tag{12}$$

**Proof.** Let $a_n = x_n - \rho_n A^*(I - T)Ax_n$ for each $n \geq 0$ and let $z \in S$. Since $\beta_n \in [0,1)$ and $\sum\limits_{n=0}^{\infty} \beta_n < \infty$, we have $\lim\limits_{n\to\infty} \beta_n = 0$. For the proof, we have the following four steps:

**Step 1.** Show that $\{x_n\}$ is bounded.
*Case $\rho_n = 0$:* Thus $a_n = x_n$. By Lemma 3, we get

$$
\begin{aligned}
\|U_{\lambda_n} a_n - z\|^2 &= \|U_{\lambda_n} x_n - z\|^2 \\
&\leq \|x_n - z\|^2 - \lambda_n(1 - \kappa - \lambda_n)\|x_n - Ux_n\|^2 \\
&\leq \|x_n - z\|^2.
\end{aligned}
$$

*Case $\rho_n \neq 0$:* By Lemmas 3 and 4, we get

$$
\begin{aligned}
\|U_{\lambda_n} a_n - z\|^2 &\leq \|a_n - z\|^2 - \lambda_n(1 - \kappa - \lambda_n)\|a_n - Ua_n\|^2 \\
&= \|x_n - \rho_n A^*(I-T)Ax_n\|^2 - \lambda_n(1 - \kappa - \lambda_n)\|a_n - Ua_n\|^2 \\
&\leq \|x_n - z\|^2 - \frac{(1-\tau)^2\|(I-T)Ax_n\|^4}{4\|A^*(I-T)Ax_n\|^2} - \lambda_n(1 - \kappa - \lambda_n)\|a_n - Ua_n\|^2 \\
&\leq \|x_n - z\|^2.
\end{aligned}
$$

Thus $\|U_\lambda a_n - z\| \leq \|x_n - z\|$. Observe that

$$
\begin{aligned}
\|x_{n+1} - z\| &= \|(1 - \beta_n)y_n + \beta_n f(y_n) - z\| \\
&\leq (1 - \beta_n)\|y_n - z\| + \beta_n\|f(y_n) - z\| \\
&\leq (1 - \beta_n)\|y_n - z\| + \beta_n\|f(y_n) - f(z)\| + \beta_n\|f(z) - z\| \\
&\leq \|y_n - z\| + \beta_n\|f(z) - z\| \\
&= \|\alpha_n f(x_n) + (1 - \alpha_n)U_{\lambda_n} a_n - z\| + \beta_n\|f(z) - z\| \\
&\leq \alpha_n\|f(x_n) - z\| + (1 - \alpha_n)\|U_{\lambda_n} a_n - z\| + \beta_n\|f(z) - z\| \\
&\leq \alpha_n\|f(x_n) - f(z)\| + \alpha_n\|f(z) - z\| \\
&\quad + (1 - \alpha_n)\|U_{\lambda_n} a_n - z\| + \beta_n\|f(z) - z\| \\
&\leq \eta\alpha_n\|x_n - z\| + \alpha_n\|f(z) - z\| + (1 - \alpha_n)\|U_{\lambda_n} a_n - z\| + \beta_n\|f(z) - z\| \\
&\leq \eta\alpha_n\|x_n - z\| + \alpha_n\|f(z) - z\| + (1 - \alpha_n)\|x_n - z\| + \beta_n\|f(z) - z\| \\
&= (1 - (1 - \eta)\alpha_n)\|x_n - z\| + \alpha_n\|f(z) - z\| + \beta_n\|f(z) - z\| \\
&\leq \max\{\|x_n - z\|, \tfrac{1}{1-\eta}\|f(z) - z\|\} + \beta_n\|f(z) - z\| \\
&\leq \max\{\|x_0 - z\|, \tfrac{1}{1-\eta}\|f(z) - z\|\} + \|f(z) - z\|\sum_{n=0}^{\infty} \beta_n.
\end{aligned}
$$

Thus $\{x_n\}$ is bounded. Moreover, $\{f(x_n)\}$, $\{y_n\}$ and $\{f(y_n)\}$ are also bounded.

**Step 2.** Show that, if the subsequence $\{x_{n_k+1}\}$ of $\{x_n\}$ weakly converges to $q \in \text{Fix}(f)$, then the subsequence $\{y_{n_k}\}$ of $\{y_n\}$ weakly converges to $q$. Now, we consider

$$
\begin{aligned}
\langle x_{n_k+1} - y_{n_k}, q \rangle &= \beta_{n_k} \langle y_{n_k} - f(y_{n_k}), q \rangle \\
&= \beta_{n_k} \frac{\|y_{n_k} - f(y_{n_k}) + q\|^2 + \|y_{n_k} - f(y_{n_k}) - q\|^2}{4}.
\end{aligned}
$$

Since $\{y_n\}$ and $\{f(y_n)\}$ are bounded, $\{y_{n_k}\}$ weakly converges to $q$.

**Step 3.** Show that the inequality holds:

$$
\begin{aligned}
\|x_{n+1} - x^*\|^2 &\leq (1 - \alpha_n)\|x_n - x^*\|^2 + 2\alpha_n \langle f(x_n) - x^*, y_n - x^* \rangle \\
&\quad + 2\beta_n \|y_n - x^*\| \|f(x^*) - x^*\| + \beta_n^2 \|f(x^*) - x^*\|^2.
\end{aligned}
$$

*Case $\rho_n = 0$:* By Lemma 3, we get

$$
\begin{aligned}
\|x_{n+1} - x^*\|^2 &= \|(1 - \beta_n)y_n + \beta_n f(y_n) - x^*\|^2 \\
&\leq ((1 - \beta_n)\|y_n - x^*\| + \beta_n \|f(y_n) - x^*\|)^2 \\
&\leq ((1 - \beta_n)\|y_n - x^*\| + \beta_n \|f(y_n) - f(x^*)\| + \beta_n \|f(x^*) - x^*\|)^2 \\
&\leq (\|y_n - x^*\| + \beta_n \|f(x^*) - x^*\|)^2 \\
&= \|y_n - x^*\|^2 + 2\beta_n \|y_n - x^*\| \|f(x^*) - x^*\| + \beta_n^2 \|f(x^*) - x^*\|^2 \\
&= \|\alpha_n f(x_n) + (1 - \alpha_n)U_{\lambda_n} x_n - x^*\|^2 \\
&\quad + 2\beta_n \|y_n - x^*\| \|f(x^*) - x^*\| + \beta_n^2 \|f(x^*) - x^*\|^2 \\
&= \|\alpha_n(f(x_n) - x^*) + (1 - \alpha_n)(U_{\lambda_n} x_n - x^*)\|^2 \\
&\quad + 2\beta_n \|y_n - x^*\| \|f(x^*) - x^*\| + \beta_n^2 \|f(x^*) - x^*\|^2 \\
&\leq (1 - \alpha_n)^2 \|U_{\lambda_n} x_n - x^*\|^2 + 2\alpha_n \langle f(x_n) - x^*, y_n - x^* \rangle \\
&\quad + 2\beta_n \|y_n - x^*\| \|f(x^*) - x^*\| + \beta_n^2 \|f(x^*) - x^*\|^2 \\
&\leq (1 - \alpha_n)^2 (\|x_n - x^*\|^2 - \lambda_n(1 - \kappa - \lambda_n)\|x_n - Ux_n\|^2) \\
&\quad + 2\alpha_n \langle f(x_n) - x^*, y_n - x^* \rangle + \beta_n^2 \|f(x^*) - x^*\|^2 \\
&\quad + 2\beta_n \|y_n - x^*\| \|f(x^*) - x^*\|.
\end{aligned}
$$

*Case $\rho_n \neq 0$:* By Lemmas 3 and 4, we get

$$
\begin{aligned}
\|x_{n+1} - x^*\|^2 &= \|(1 - \beta_n)y_n + \beta_n f(y_n) - x^*\|^2 \\
&\leq ((1 - \beta_n)\|y_n - x^*\| + \beta_n \|f(y_n) - x^*\|)^2 \\
&\leq ((1 - \beta_n)\|y_n - x^*\| + \beta_n \|f(y_n) - f(x^*)\| + \beta_n \|f(x^*) - x^*\|)^2 \\
&\leq \|y_n - x^*\|^2 + 2\beta_n \|y_n - x^*\| \|f(x^*) - x^*\| + \beta_n^2 \|f(x^*) - x^*\|^2 \\
&= \|\alpha_n f(x_n) + (1 - \alpha_n)U_{\lambda_n} a_n - x^*\|^2 \\
&\quad + 2\beta_n \|y_n - x^*\| \|f(x^*) - x^*\| + \beta_n^2 \|f(x^*) - x^*\|^2 \\
&= \|\alpha_n(f(x_n) - x^*) + (1 - \alpha_n)(U_{\lambda_n} a_n - x^*)\|^2 \\
&\quad + 2\beta_n \|y_n - x^*\| \|f(x^*) - x^*\| + \beta_n^2 \|f(x^*) - x^*\|^2 \\
&\leq (1 - \alpha_n)^2 \|U_{\lambda_n} a_n - x^*\|^2 + 2\alpha_n \langle f(x_n) - x^*, y_n - x^* \rangle \\
&\quad + 2\beta_n \|y_n - x^*\| \|f(x^*) - x^*\| + \beta_n^2 \|f(x^*) - x^*\|^2 \\
&\leq (1 - \alpha)^2 (\|a_n - x^*\|^2 - \lambda_n(1 - \kappa - \lambda_n)\|a_n - Ua_n\|^2) \\
&\quad + 2\alpha_n \langle f(x_n) - x^*, y_n - x^* \rangle + \beta_n^2 \|f(x^*) - x^*\|^2 \\
&\quad + 2\beta_n \|y_n - x^*\| \|f(x^*) - x^*\| \\
&\leq (1 - \alpha)^2 (\|x_n - x^*\|^2 - \frac{(1 - \tau)^2 \|(I - T)Ax_n\|^4}{4\|A^*(I - T)Ax_n\|^2} \\
&\quad - \lambda_n(1 - \kappa - \lambda_n)\|a_n - Ua_n\|^2) \\
&\quad + 2\alpha_n \langle f(x_n) - x^*, y_n - x^* \rangle + \beta_n^2 \|f(x^*) - x^*\|^2 \\
&\quad + 2\beta_n \|y_n - x^*\| \|f(x^*) - x^*\|.
\end{aligned}
$$

Therefore, we have

$$\|x_{n+1} - x^*\|^2 \leq (1 - \alpha_n)\|x_n - x^*\|^2 + 2\alpha_n\langle f(x_n) - x^*, y_n - x^*\rangle$$
$$+ 2\beta_n\|y_n - x^*\|\|f(x^*) - x^*\| + \beta_n^2\|f(x^*) - x^*\|^2.$$

**Step 4.** Show that $x_n \to x^*$ for each $n \geq 0$. Let $s_n = \|x_n - x^*\|$. In this step, we consider two cases.

*Case 1.* Assume that there is $n_0 \in \mathbb{N}$ such that $\{s_n\}$ is decreasing for all $n \geq n_0$. Since $\{s_n\}$ is monotonic and bounded, $\{s_n\}$ is convergent. First, we show that

$$\limsup_{n\to\infty}\langle f(x^*) - x^*, y_n - x^*\rangle \leq 0.$$

There are two parts to show this.

*Part 1.* Let $\rho_n = 0$. Since $\{f(x_n)\}$ and $\{y_n\}$ are bounded and Step 3, we get

$$\lambda_n(1 - \kappa - \lambda_n)\|x_n - Ux_n\|^2 \leq s_n - s_{n+1} + \alpha_n M + \beta_n N,$$

where

$$M = \sup_{n\in\mathbb{N}}\{2\langle f(x_n) - x^*, y_n - x^*\rangle\}$$

and

$$N = \sup_{n\in\mathbb{N}}\{2\|y_n - x^*\|\|f(x^*) - x^*\| + \beta_n\|f(x^*) - x^*\|^2\}.$$

Since $\{s_n\}$ is convergent and $\lim_{n\to\infty}\alpha_n = 0$, we have $\lim_{n\to\infty}\|x_n - Ux_n\| = 0$. By since $\rho_n = 0$, we have

$$\lim_{n\to\infty}\|(I - T)Ax_n\| = 0.$$

By the boundedness of $\{x_n\}$, there is a subsequence $\{x_{n_k}\}$ of $\{x_n\}$ such that $x_{n_k} \rightharpoonup q$ and

$$\limsup_{n\to\infty}\langle f(x^*) - x^*, x_n - x^*\rangle = \lim_{k\to\infty}\langle f(x^*) - x^*, x_{n_k} - x^*\rangle$$
$$= \langle f(x^*) - x^*, q - x^*\rangle.$$

Since $\lim_{n\to\infty}\|x_n - Ux_n\| = 0$ and the demiclosedness of $I - U$ at zero, we have $q \in \text{Fix}(U)$. Since $A$ is a bounded linear operator, $A$ is continuous. Therefore, $x_{n_k} \rightharpoonup q$ imply $Ax_{n_k} \rightharpoonup Aq$. Form $\lim_{n\to\infty}\|(I - T)Ax_n\| = 0$ and the demiclosedness of $I - T$ at zero, it follows that $Aq \in \text{Fix}(T)$ and so $q \in S$. By Step 2, it follows that

$$0 \geq \langle f(x^*) - x^*, q - x^*\rangle = \lim_{k\to\infty}\langle f(x^*) - x^*, x_{n_k} - x^*\rangle$$
$$= \limsup_{n\to\infty}\langle f(x^*) - x^*, x_n - x^*\rangle$$
$$= \limsup_{n\to\infty}\langle f(x^*) - x^*, y_{n-1} - x^*\rangle.$$

*Part 2.* Let $\rho_n \neq 0$. Since $\{f(x_n)\}$ and $\{y_n\}$ are bounded, by Step 3, we get

$$\lambda_n(1 - \kappa - \lambda_n)\|a_n - Ua_n\|^2 + \frac{(1 - \tau)^2\|(I - T)Ax_n\|^4}{4\|A^*(I - T)Ax_n\|^2} \leq s_n - s_{n+1} + \alpha_n M + \beta_n N,$$

where

$$M = \sup_{n\in\mathbb{N}}\{2\langle f(x_n) - x^*, y_n - x^*\rangle\}$$

and

$$N = \sup_{n \in \mathbb{N}} \{ 2\|y_n - x^*\| \|f(x^*) - x^*\| + \beta_n \|f(x^*) - x^*\|^2 \}.$$

Thus we obtain

$$0 \le \lambda_n (1 - \kappa - \lambda_n) \|a_n - Ua_n\|^2 \le s_n - s_{n+1} + \alpha_n M + \beta_n N$$

and

$$0 \le \frac{(1-\tau)^2 \|(I-T)Ax_n\|^4}{4\|A^*(I-T)Ax_n\|^2} \le s_n - s_{n+1} + \alpha_n M + \beta_n N.$$

Since $\{s_n\}$ is convergent and $\lim_{n \to \infty} \alpha_n = 0$, we obtain

$$\lim_{n \to \infty} \|a_n - Ua_n\| = \lim_{n \to \infty} \frac{\|(I-T)Ax_n\|^4}{\|A^*(I-T)Ax_n\|^2} = 0.$$

Moreover, we get $\lim_{n \to \infty} \|(I - T)Ax_n\| = 0$. However, it follows that

$$\|(I-T)Ax_n\| = \|A\| \|(I-T)Ax_n\|^2 \frac{1}{\|A\| \|(I-T)Ax_n\|} \le \|A\| \frac{\|(I-T)Ax_n\|^2}{\|A^*(I-T)Ax_n\|}.$$

Thus we have

$$\lim_{n \to \infty} \|x_n - a_n\| = \lim_{n \to \infty} \frac{(1-\tau)\|(I-T)Ax_n\|^2}{2\|A^*(I-T)Ax_n\|} = 0.$$

By the boundedness of $\{x_n\}$, there is a subsequence $\{x_{n_k}\}$ of $\{x_n\}$ such that $x_{n_k} \rightharpoonup q$. Since $\lim_{n \to \infty} \|x_n - a_n\| = 0$ and $x_{n_k} \rightharpoonup q$, there is a subsequence $\{a_{n_k}\}$ of $\{a_n\}$ such that $a_{n_k} \rightharpoonup q$ and

$$\limsup_{n \to \infty} \langle f(x^*) - x^*, x_n - x^* \rangle = \lim_{k \to \infty} \langle f(x^*) - x^*, x_{n_k} - x^* \rangle = \langle f(x^*) - x^*, q - x^* \rangle.$$

Since $\lim_{n \to \infty} \|a_n - Ua_n\| = 0$, by the demiclosedness of $I - U$ at zero, we have $q \in \text{Fix}(U)$. Since $A$ is a bounded linear operator, $A$ is continuous. Therefore, $x_{n_k} \rightharpoonup q$ imply $Ax_{n_k} \rightharpoonup Aq$. Form $\lim_{n \to \infty} \|(I - T)Ax_n\| = 0$ and the demiclosedness of $I - T$ at zero, we have $Aq \in \text{Fix}(T)$ and $q \in S$. By Step 2, it follow that

$$\begin{aligned} 0 &\ge \langle f(x^*) - x^*, q - x^* \rangle = \lim_{k \to \infty} \langle f(x^*) - x^*, x_{n_k} - x^* \rangle \\ &= \limsup_{n \to \infty} \langle f(x^*) - x^*, x_n - x^* \rangle \\ &= \limsup_{n \to \infty} \langle f(x^*) - x^*, y_{n-1} - x^* \rangle. \end{aligned}$$

Second, we show that $\lim_{n \to \infty} \|x_{n+1} - x_n\| = 0$. There are two parts.

*Part 1.* If $\rho_n = 0$, then we get

$$\begin{aligned} \|x_{n+1} - x_n\| &= \|(1 - \beta_n)y_n + \beta_n f(y_n) - x_n\| \\ &\le (1 - \beta_n)\|y_n - x_n\| + \beta_n \|f(y_n) - x_n\| \\ &= (1 - \beta_n)\|\alpha_n f(x_n) + (1 - \alpha_n)U_{\lambda_n} x_n - x_n\| + \beta_n \|f(y_n) - x_n\| \\ &\le \alpha_n \|f(x_n) - x_n\| + (1 - \alpha_n)\|x_n - U_{\lambda_n} x_n\| + \beta_n \|f(y_n) - x_n\| \\ &= \alpha_n \|f(x_n) - x_n\| + \lambda_n \|x_n - Ux_n\| + \beta_n \|f(y_n) - x_n\|. \end{aligned}$$

*Part 2.* If $\rho_n \neq 0$, then we get

$$
\begin{aligned}
\|x_{n+1} - x_n\| &\leq \|(1 - \beta_n)y_n + \beta_n f(y_n) - x_n\| \\
&\leq (1 - \beta_n)\|y_n - x_n\| + \beta_n\|f(y_n) - x_n\| \\
&= (1 - \beta_n)\|\alpha_n f(x_n) + (1 - \alpha_n)U_{\lambda_n}a_n - x_n\| + \beta_n\|f(y_n) - x_n\| \\
&\leq \alpha_n\|f(x_n) - x_n\| + (1 - \alpha_n)\|x_n - U_{\lambda_n}a_n\| + \beta_n\|f(y_n) - x_n\| \\
&\leq \alpha_n\|f(x_n) - x_n\| + \|x_n - a_n\| + \|a_n - U_{\lambda_n}a_n\| + \beta_n\|f(y_n) - x_n\| \\
&= \alpha_n\|f(x_n) - x_n\| + \|x_n - a_n\| + \lambda_n\|a_n - Ua_n\| + \beta_n\|f(y_n) - x_n\|.
\end{aligned}
$$

Therefore, we have $\lim_{n \to \infty} \|x_{n+1} - x_n\| = 0$.

Third, we show that $x_n \to x^*$. We get the inequality:

$$
\limsup_{n \to \infty} \langle f(x^*) - x^*, y_n - x^* \rangle \leq 0.
$$

Now, we have

$$
\begin{aligned}
\|x_{n+1} - x^*\|^2 &\leq (1 - \alpha_n)\|x_n - x^*\|^2 + 2\alpha_n \limsup_{k \to \infty} \langle f(x_k) - x^*, y_k - x^* \rangle \\
&\quad + \beta_n \sup_{k \in \mathbb{N}} \{2\|y_k - x^*\|\|f(x^*) - x^*\| + \beta_k\|f(x^*) - x^*\|^2\}.
\end{aligned}
$$

By Lemma 2, we have $\lim_{n \to \infty} s_n = \lim_{n \to \infty} \|x_n - x^*\| = 0$ and so $x_n \to x^*$.

*Case 2.* Assume that there is not $n_0 \in \mathbb{N}$ such that $\{s_n\}$ is decreasing for all $n \geq n_0$. Thus there is a subsequence $\{s_{n_i+1}\}$ of $\{s_n\}$ such that $s_{n_i+1} < s_{n_{i+1}+1}$ for all $i \in \mathbb{N}$.

First, we show that

$$
\limsup_{n_i \to \infty} \langle f(x^*) - x^*, y_{n_i} - x^* \rangle \leq 0.
$$

There are two parts.

*Part 1.* Let $\rho_{n_i} = 0$. Since $\{f(x_{n_i})\}$ and $\{y_{n_i}\}$ are bounded, by Step 3, we get

$$
\lambda_{n_i}(1 - \kappa - \lambda_{n_i})\|x_{n_i} - Ux_{n_i}\|^2 \leq s_{n_i} - s_{n_i+1} + \alpha_{n_i}M + \beta_{n_i}N \leq \alpha_{n_i}M + \beta_{n_i}N,
$$

where

$$
M \in \mathbb{R} = \sup_{n_i \in \mathbb{N}} \{2\langle f(x_{n_i}) - x^*, y_{n_i} - x^* \rangle\}
$$

and

$$
N = \sup_{n \in \mathbb{N}} \{2\|y_{n_i} - x^*\|\|f(x^*) - x^*\| + \beta_{n_i}\|f(x^*) - x^*\|^2\}.
$$

Since $\lim_{i \to \infty} \alpha_{n_i} = 0$, we have

$$
\lim_{i \to \infty} \|x_{n_i} - Ux_{n_i}\| = 0.
$$

Since $\rho_{n_i} = 0$, we have

$$
\lim_{i \to \infty} \|(I - TA)x_{n_i}\| = 0.
$$

By the boundedness of $\{x_{n_i}\}$, there is a subsequence $\{x_{n_{i_j}}\}$ of $\{x_{n_i}\}$ such that $x_{n_{i_j}} \rightharpoonup q$ and

$$
\limsup_{i \to \infty} \langle f(x^*) - x^*, x_{n_i} - x^* \rangle = \lim_{j \to \infty} \langle f(x^*) - x^*, x_{n_{i_j}} - x^* \rangle = \langle f(x^*) - x^*, q - x^* \rangle.
$$

Since $\lim_{j \to \infty} \|x_{n_{i_j}} - Ux_{n_{i_j}}\| = 0$ and the demiclosedness of $I - U$ at zero, we have $q \in \text{Fix}(U)$. Since $A$ is a bounded linear operator, $A$ is continuous. Therefore, $x_{n_{i_j}} \rightharpoonup q$ imply $Ax_{n_{i_j}} \rightharpoonup Aq$. Form

$\lim\limits_{j\to\infty}\|(I-T)Ax_{n_{i_j}}\|=0$ and the demiclosedness of $I-T$ at zero, we have $Aq\in\text{Fix}(T)$ and so $q\in S$. By Step 2, it follows that

$$
\begin{aligned}
0 \;&\geq\; \langle f(x^*)-x^*, q-x^*\rangle = \lim_{j\to\infty}\langle f(x^*)-x^*, x_{n_{i_j}}-x^*\rangle \\
&=\; \limsup_{i\to\infty}\langle f(x^*)-x^*, x_{n_i}-x^*\rangle \\
&=\; \limsup_{i\to\infty}\langle f(x^*)-x^*, y_{n_i-1}-x^*\rangle.
\end{aligned}
$$

*Part 2.* Let $\rho_{n_i}\neq 0$. Since $\{f(x_{n_i})\}$ and $\{y_{n_i}\}$ are bounded, by Step 3, we get

$$
\begin{aligned}
&\lambda_{n_i}(1-\kappa-\lambda_{n_i})\|a_{n_i}-Ua_{n_i}\|^2 + \frac{(1-\tau)^2\|(I-T)Ax_{n_i}\|^4}{4\|A^*(I-T)Ax_{n_i}\|^2} \\
&\qquad \leq s_{n_i}-s_{n_{i+1}}+\alpha_{n_i}M+\beta_{n_i}N \\
&\qquad \leq \alpha_{n_i}M+\beta_{n_i}N,
\end{aligned}
$$

where

$$
M=\sup_{n_i\in\mathbb{N}}\{2\langle f(x_{n_i})-x^*, y_{n_i}-x^*\rangle\}
$$

and

$$
N=\sup_{n\in\mathbb{N}}\{2\|y_{n_i}-x^*\|\,\|f(x^*)-x^*\|+\beta_{n_i}\|f(x^*)-x^*\|^2\}.
$$

Then we obtain

$$
0\leq \lambda_{n_i}(1-\kappa-\lambda_{n_i})\|a_{n_i}-Ua_{n_i}\|^2\leq \alpha_{n_i}M+\beta_{n_i}N
$$

and

$$
0\leq \frac{(1-\tau)^2\|(I-T)Ax_{n_i}\|^4}{4\|A^*(I-T)Ax_{n_i}\|^2}\leq \alpha_{n_i}M+\beta_{n_i}N.
$$

Since $\lim\limits_{i\to\infty}\alpha_{n_i}=0$, we obtain

$$
\lim_{i\to\infty}\|a_{n_i}-Ua_{n_i}\|=\lim_{i\to\infty}\frac{\|(I-T)Ax_{n_i}\|^4}{\|A^*(I-T)Ax_{n_i}\|^2}=0.
$$

Moreover, we get $\lim\limits_{n\to\infty}\|(I-T)Ax_{n_i}\|=0$. However, we have

$$
\|(I-T)Ax_{n_i}\|=\|A\|\,\|(I-T)Ax_{n_i}\|^2\frac{1}{\|A\|\,\|(I-T)Ax_{n_i}\|}\leq\|A\|\frac{\|(I-T)Ax_{n_i}\|^2}{\|A^*(I-T)Ax_{n_i}\|}.
$$

Thus we have

$$
\lim_{i\to\infty}\|x_{n_{i+1}}-a_{n_i}\|=\lim_{i\to\infty}\frac{(1-\tau)\|(I-T)Ax_{n_i}\|^2}{2\|A^*(I-T)Ax_{n_i}\|}=0.
$$

By the boundedness of $\{x_{n_i}\}$, there is a subsequence $\{x_{n_{i_j}}\}$ of $\{x_{n_i}\}$ and $x_{n_{i_j}}\rightharpoonup q$. Since $\lim\limits_{i\to\infty}\|x_{n_i}-a_{n_i}\|=0$ and $x_{n_{i_j}}\rightharpoonup q$, we have $a_{n_{i_j}}\rightharpoonup q$ such that

$$
\limsup_{i\to\infty}\langle f(x^*)-x^*, x_{n_i}-x^*\rangle=\lim_{j\to\infty}\langle f(x^*)-x^*, x_{n_{i_j}}-x^*\rangle=\langle f(x^*)-x^*, q-x^*\rangle.
$$

Since $\lim\limits_{i\to\infty}\|a_{n_i}-Ua_{n_i}\|=0$, by the demiclosedness of $I-U$ at zero, we have $q\in\text{Fix}(U)$. Since $A$ is a bounded linear operator, $A$ is continuous. Therefore, $x_{n_{i_j}}\rightharpoonup q$ imply $Ax_{n_{i_j}}\rightharpoonup Aq$. Form $\lim\limits_{i\to\infty}\|(I-$

$T)Ax_{n_i}\| = 0$ and the demiclosedness of $I - T$ at zero, we have $Aq \in \text{Fix}(T)$ and so $q \in S$. By Step 2, it follows that

$$
\begin{aligned}
0 &\geq \langle f(x^*) - x^*, q - x^* \rangle = \lim_{j \to \infty} \langle f(x^*) - x^*, x_{n_{i_j}} - x^* \rangle \\
&= \limsup_{i \to \infty} \langle f(x^*) - x^*, x_{n_i} - x^* \rangle \\
&= \limsup_{i \to \infty} \langle f(x^*) - x^*, y_{n_i-1} - x^* \rangle.
\end{aligned}
$$

Second, we show that

$$
\lim_{i \to \infty} \|x_{n_{i+1}} - x_{n_i}\| = 0.
$$

There are two parts.

*Part 1.* If $\rho_{n_i} = 0$, then we compute

$$
\begin{aligned}
\|x_{n_{i+1}} - x_{n_i}\| &\leq \|(1 - \beta_{n_i})y_{n_i} + \beta_{n_i}f(y_{n_i}) - x_{n_i}\| \\
&\leq (1 - \beta_{n_i})\|y_{n_i} - x_{n_i}\| + \beta_{n_i}\|f(y_{n_i}) - x_{n_i}\| \\
&= (1 - \beta_{n_i})\|\alpha_{n_i}f(x_{n_i}) + (1 - \alpha_{n_i})U_{\lambda_{n_i}}x_{n_i} - x_{n_i}\| + \beta_{n_i}\|f(y_{n_i}) - x_{n_i}\| \\
&\leq \alpha_{n_i}\|f(x_{n_i}) - x_{n_i}\| + (1 - \alpha_{n_i})\|x_{n_i} - U_{\lambda_{n_i}}x_{n_i}\| + \beta_{n_i}\|f(y_{n_i}) - x_{n_i}\| \\
&= \alpha_{n_i}\|f(x_{n_i}) - x_{n_i}\| + \lambda_{n_i}\|x_{n_i} - Ux_{n_i}\| + \beta_{n_i}\|f(y_{n_i}) - x_{n_i}\|.
\end{aligned}
$$

*Part 2.* If $\rho_{n_i} \neq 0$, then we compute

$$
\begin{aligned}
\|x_{n_{i+1}} - x_{n_i}\| &\leq \|(1 - \beta_{n_i})y_{n_i} + \beta_{n_i}f(y_{n_i}) - x_{n_i}\| \\
&\leq (1 - \beta_{n_i})\|y_{n_i} - x_{n_i}\| + \beta_{n_i}\|f(y_{n_i}) - x_{n_i}\| \\
&= (1 - \beta_{n_i})\|\alpha_{n_i}f(x_{n_i}) + (1 - \alpha_{n_i})U_{\lambda_{n_i}}a_{n_i} - x_{n_i}\| + \beta_{n_i}\|f(y_{n_i}) - x_{n_i}\| \\
&\leq \alpha_{n_i}\|f(x_{n_i}) - x_{n_i}\| + (1 - \alpha_{n_i})\|x_{n_i} - U_{\lambda_{n_i}}a_{n_i}\| + \beta_{n_i}\|f(y_{n_i}) - x_{n_i}\| \\
&\leq \alpha_{n_i}\|f(x_{n_i}) - x_{n_i}\| + \|x_{n_i} - a_{n_i}\| + \|a_{n_i} - U_{\lambda_n}a_{n_i}\| + \beta_{n_i}\|f(y_{n_i}) - x_{n_i}\| \\
&= \alpha_{n_i}\|f(x_{n_i}) - x_{n_i}\| + \|x_{n_i} - a_{n_i}\| + \lambda_{n_i}\|a_{n_i} - Ua_{n_i}\| + \beta_{n_i}\|f(y_{n_i}) - x_{n_i}\|.
\end{aligned}
$$

Therefore, we have

$$
\lim_{i \to \infty} \|x_{n_{i+1}} - x_{n_i}\| = 0.
$$

Third, we show that $x_n \to x^*$. From the inequality $s_{n_i+1} \leq s_{n_{i+1}+1}$, we get

$$
\limsup_{i \to \infty} \langle f(x^*) - x^*, y_{n_i} - x^* \rangle \leq 0.
$$

Observe that

$$
\begin{aligned}
\alpha_{n_i}s_{n_i+1} + (1 - \alpha_{n_i})(s_{n_i+1} - s_{n_i}) &\leq 2\alpha_{n_i}\limsup_{i \to \infty}\langle f(x^*) - x^*, y_{n_i} - x^* \rangle \\
&\quad + \beta_{n_i}\sup_{k \in \mathbb{N}}\{2\|y_k - x^*\|\|f(x^*) - x^*\| + \beta_k\|f(x^*) - x^*\|^2\}.
\end{aligned}
$$

Then we have

$$
\begin{aligned}
0 \leq s_{n_i+1} &\leq 2\limsup_{i \to \infty}\langle f(x^*) - x^*, y_{n_i} - x^* \rangle \\
&\quad + \beta_{n_i}\sup_{k \in \mathbb{N}}\{2\|y_k - x^*\|\|f(x^*) - x^*\| + \beta_k\|f(x^*) - x^*\|^2\}.
\end{aligned}
$$

Therefore, since $\{y_n\}$ is bounded and $\lim_{n \to \infty} \beta_n = 0$, from $\lim_{n \to \infty} s_n = \lim_{n \to \infty} \|x_n - x^*\| = 0$, it follows that $x_n \to x^*$. This completes the proof. $\square$

## 4. Special Cases

We consider some special cases of Theorem 1 based on some relations of directed operators, $\tau$-demicontractive operators and quasi-nonexpansive operators. See Figure 1. For some details, see Remark 2. Therefore, the following results follows easily from Theorem 1:

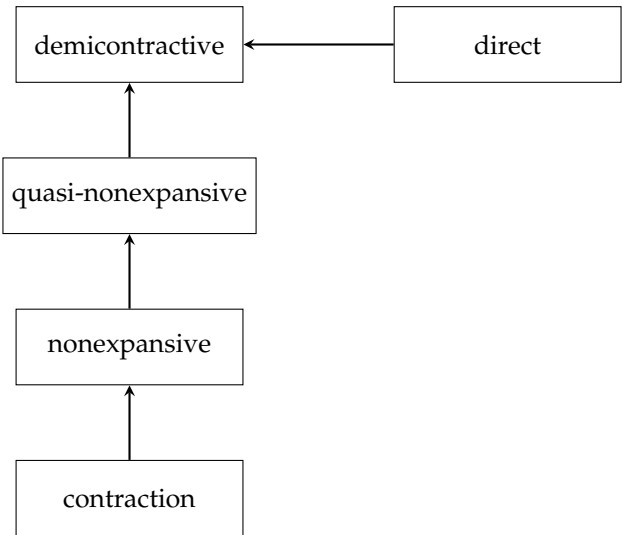

**Figure 1.** Diagram relations operator.

*Case 1.* Assume that $U : H \to H$ is a quasi-nonexpansive operator such that $I - U$ is demiclosed at zero and $T : K \to K$ is a quasi-nonexpansive operator such that $I - T$ is demiclosed at zero, respectively.

**Corollary 1.** *Assume that $S$ is a set of all solutions of the problem (6) such that $S \neq \varnothing$. Suppose that*

$$\sum_{n=0}^{\infty} \beta_n < \infty, \quad \lim_{n \to \infty} \alpha_n = 0, \quad \sum_{n=0}^{\infty} \alpha_n = \infty.$$

*Then the sequence $\{x_n\}$ generated by the algorithm (11) converges strongly to $x^* \in S$ and, also, $x^* = P_S f(x^*)$ is a solution of the variational inequality (12).*

*Case 2.* Assume that $U : H \to H$ is a quasi-nonexpansive operator such that $I - U$ is demiclosed at zero and $T : K \to K$ is a directed operator such that $I - T$ is demiclosed at zero, respectively.

**Corollary 2.** *Assume that $S$ is a set of all solutions of the problem (6) such that $S \neq \varnothing$. Suppose that*

$$\sum_{n=0}^{\infty} \beta_n < \infty, \quad \lim_{n \to \infty} \alpha_n = 0, \quad \sum_{n=0}^{\infty} \alpha_n = \infty$$

*Then the sequence $\{x_n\}$ generated by the algorithm (11) converges strongly to $x^* \in S$ and, also, $x^* = P_S f(x^*)$ is a solution of the variational inequality (12).*

*Case 3.* Assume that $U : H \to H$ is a directed operator such that $I - U$ is demiclosed at zero and $T : K \to K$ is a quasi-nonexpansive operator such that $I - T$ is demiclosed at zero, respectively.

**Corollary 3.** *Assume that $S$ is a set of all solutions of the problem (6) such that $S \neq \varnothing$. Suppose that*

$$\sum_{n=0}^{\infty} \beta_n < \infty, \quad \lim_{n \to \infty} \alpha_n = 0, \quad \sum_{n=0}^{\infty} \alpha_n = \infty.$$

Then the sequence $\{x_n\}$ generated by the algorithm (11) converges strongly to $x^* \in S$ and, also, $x^* = P_S f(x^*)$ is a solution of the variational inequality (12).

*Case 4.* Assume that $U : H \to H$ is a quasi-nonexpansive operator such that $I - U$ is demiclosed at zero and $T : K \to K$ is a $\tau$-demicontractive operator such that $I - T$ is demiclosed at zero, respectively.

**Corollary 4.** *Assume that $S$ is a set of all solutions of the problem (6) such that $S \neq \emptyset$. Suppose that*

$$\sum_{n=0}^{\infty} \beta_n < \infty, \quad \lim_{n \to \infty} \alpha_n = 0, \quad \sum_{n=0}^{\infty} \alpha_n = \infty.$$

*Then the sequence $\{x_n\}$ generated by the algorithm (11) converges strongly to $x^* \in S$ and, also, $x^* = P_S f(x^*)$ is a solution of the variational inequality (12).*

*Case 5.* Assume that $U : H \to H$ is a $\tau$-demicontractive operator such that $I - U$ is demiclosed at zero and $T : K \to K$ is a quasi-nonexpansive operator such that $I - T$ is demiclosed at zero, respectively.

**Corollary 5.** *Assume that $S$ is a set of all solutions of the problem (6) such that $S \neq \emptyset$. Suppose that*

$$\sum_{n=0}^{\infty} \beta_n < \infty, \quad \lim_{n \to \infty} \alpha_n = 0, \quad \sum_{n=0}^{\infty} \alpha_n = \infty.$$

*Then the sequence $\{x_n\}$ generated by the algorithm (11) converges strongly to $x^* \in S$ and, also, $x^* = P_S f(x^*)$ is a solution of the variational inequality (12).*

*Case 6.* Assume that $U : H \to H$ is a directed operator such that $I - U$ is demiclosed at zero and $T : K \to K$ is a directed operator such that $I - T$ is demiclosed at zero, respectively.

**Corollary 6.** *Assume that $S$ is a set of all solutions of the problem (6) such that $S \neq \emptyset$. Suppose that*

$$\sum_{n=0}^{\infty} \beta_n < \infty, \quad \lim_{n \to \infty} \alpha_n = 0, \quad \sum_{n=0}^{\infty} \alpha_n = \infty.$$

*Then the sequence $\{x_n\}$ generated by the algorithm (11) converges strongly to $x^* \in S$ and, also, $x^* = P_S f(x^*)$ is a solution of the variational inequality (12).*

*Case 7.* Assume that $U : H \to H$ is a directed operator such that $I - U$ is demiclosed at zero and $T : K \to K$ is a $\tau$-demicontractive operator such that $I - T$ is demiclosed at zero, respectively.

**Corollary 7.** *Assume that $S$ is a set of all solutions of the problem (6) such that $S \neq \emptyset$. Suppose that*

$$\sum_{n=0}^{\infty} \beta_n < \infty, \quad \lim_{n \to \infty} \alpha_n = 0, \quad \sum_{n=0}^{\infty} \alpha_n = \infty.$$

*Then the sequence $\{x_n\}$ generated by the algorithm (11) converges strongly to $x^* \in S$ and, also, $x^* = P_S f(x^*)$ is a solution the variational inequality (12).*

*Case 8.* Assume that $U : H \to H$ is a $\tau$-demicontractive operator such that $I - U$ is demiclosed at zero and $T : K \to K$ is a directed operator such that $I - T$ is demiclosed at zero, respectively.

**Corollary 8.** *Assume that S is a set of all solutions of the problem* (6) *such that $S \neq \emptyset$. Suppose that*

$$\sum_{n=0}^{\infty} \beta_n < \infty, \quad \lim_{n \to \infty} \alpha_n = 0, \quad \sum_{n=0}^{\infty} \alpha_n = \infty.$$

*Then the sequence $\{x_n\}$ generated by the algorithm* (11) *converges strongly to $x^* \in S$ and, also, $x^* = P_S f(x^*)$ is a solution of the variational inequality* (12).

## 5. Application to Signal Processing

For most of the contents in this section, we follow those of Cui and Ceng [21]. We consider some applications of our algorithm to inverse problems occurring from signal processing. For example, we consider the following equation:

$$y = Ax + \epsilon, \tag{13}$$

where $x \in \mathbb{R}^N$ is recovered, $y \in \mathbb{R}^k$ is noisy observations, $A : \mathbb{R}^N \to \mathbb{R}^k$ is a bounded linear observation operator. It determines a process with loss of information. For finding solutions of the linear inverse problems (13), a successful one of some models is the convex unconstrained minimization problem:

$$\min_{x \in \mathbb{R}^N} \frac{1}{2} \|y - Ax\|^2 + v\|x\|_1, \tag{14}$$

where $v > 0$ and $\|\cdot\|_1$ is the $\ell_1$ norm. It is well know that the problem (14) is equivalent to the constrained least squares problem:

$$\min_{x \in \mathbb{R}^N} \frac{1}{2} \|y - Ax\|^2 \text{ subject to } x \in C, \tag{15}$$

where $C = \{x \in \mathbb{R}^N : \|x\|_1 \leq t\}$. The problem (15) is a particular case of the problem (1), where $Q = \{y\}$. Therefore, we can solve the problem by the proposed algorithm. In this case, $U = P_C$ is the projection onto the closed $\ell_1$-ball in $\mathbb{R}^N$ and $T = P_Q$, see [22,23]. Denoted $P_{C_{\lambda_n}} := (1 - \lambda_n)I + \lambda_n P_C$ for each $n \geq 1$, where $\lambda_n \in (0, 1)$. Then we have the following algorithm:

$$\begin{cases} y_n = \alpha_n f(x_n) + (1 - \alpha_n)P_{C_{\lambda_n}}(x_n - \rho_n A^*(I - P_Q)Ax_n), \\ x_{n+1} = (1 - \beta_n)y_n + \beta_n f(y_n), \quad \forall n \geq 0, \end{cases} \tag{16}$$

where

$$\rho_n = \begin{cases} \frac{(1-\tau)\|(Ax_n - y)\|^2}{2\|A^*(Ax_n - y)\|^2}, & Ax_n \neq y, \\ 0 & \text{otherwise,} \end{cases}$$

$f$ is a contraction operator on $C$ and the sequences $\{\alpha_n\}$, $\{\beta_n\}$ in $[0, 1)$ are such that

$$\lim_{n \to \infty} \alpha_n = 0, \quad \sum_{n=0}^{\infty} \alpha_n = \infty, \quad \sum_{n=0}^{\infty} \beta_n < \infty.$$

**Theorem 2.** *Then the sequence $\{x_n\}$ generated by the algorithm* (16) *converges strongly to a solution $x^*$ of the problem* (15).

**Example 1.** *Let A be the random matrix $(k \times N)$ such that each entire is in $[0, 1]$. Let $y = Ax^*$ be such that $\|x^*\|_1 \leq t$. Set up the problem* (15). *We choose $\lambda = \frac{1}{2}, \alpha = \frac{1}{n}, \beta = \frac{1}{n^2}, u = \begin{bmatrix} 1 & \cdots & 1 \end{bmatrix}^\dagger, f(x) = (x - p)/4 + p$ and initial $x_1$ randomly be such that $\|p\|_1, \|x_1\|_1 \leq t$. Thus $C = \{x \in \mathbb{R}^N : \|x\|_1 \leq t\}$. See Figures 2 and 3.*

**Remark 4.** *Figures 2–5 show that the sequence $\{\beta_n\}$ improves the convergence profile of [14,15]. Our algorithm (Algorithm 5) converges faster than Cui and Wang's algorithm and Boikanyo's algorithm. Moreover, we compared our algorithm with the forward-backward splitting algorithm [24] and the fast iterative shrinkage-thresholding algorithm (FISTA) [25]. Sometimes, our algorithm converges faster than other algorithms, Figures 4 and 5, but, sometimes, our algorithm converges slower than other algorithms, Figures 2 and 3 . It depends on the control condition. This experiment is an example for the convergence of some algorithms.*

---

**Algorithm 5:** A General Viscosity Algorithms (Our Algorithm)

---

**Input:** Set $\lambda_n \in (0,1)$, $\alpha_n$, $\beta_n \in [0,1)$ such that $\lim\limits_{n\to\infty} \alpha_n = 0$, $\sum\limits_{n=0}^{\infty} \alpha_n = \infty$, $\sum\limits_{n=0}^{\infty} \beta_n < \infty$. Choose
$\quad x_0 \in H$.

1 **for** $n = 1, 2, \cdots$ **do**

2     **if** $Ax_n \neq y$, **then**

3         $\rho_n = \dfrac{(1-\tau)\|(Ax_n - y)\|^2}{2\|A^*(Ax_n - y)\|^2}$

4     **else**

5         $\rho_n = 0$

6     **end**

7     $y_n = \alpha_n f(x_n) + (1-\alpha_n)P_{C_{\lambda_n}}(x_n - \rho_n A^*(Ax_n - y))$

8     $x_{n+1} = (1 - \beta_n)y_n + \beta_n f(y_n)$

9 **end for**

---

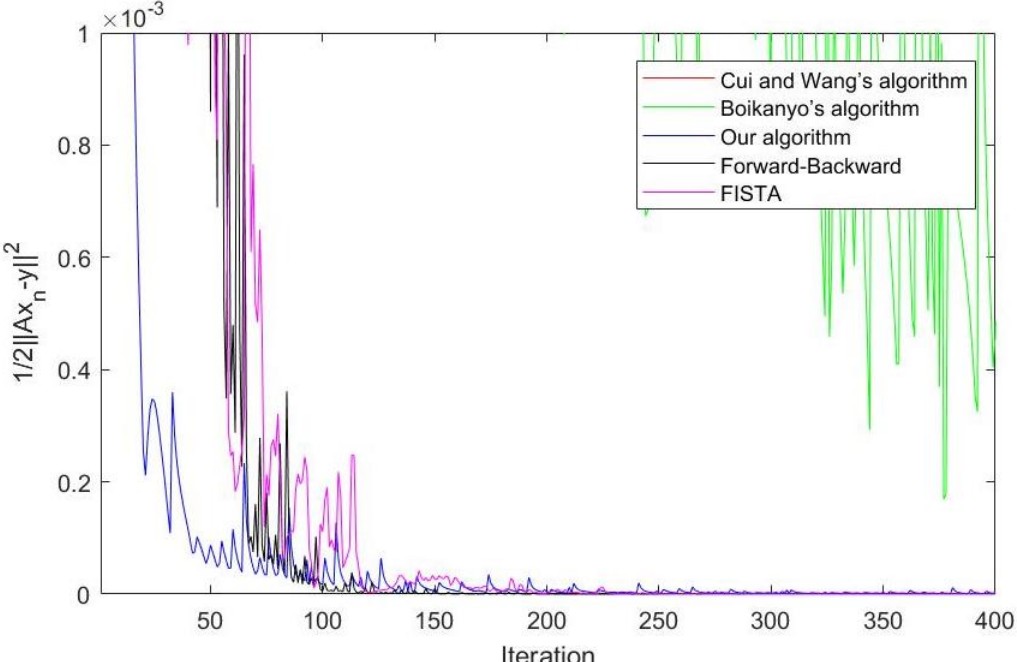

**Figure 2.** Case $N = t = 10$ and $k = 9$.

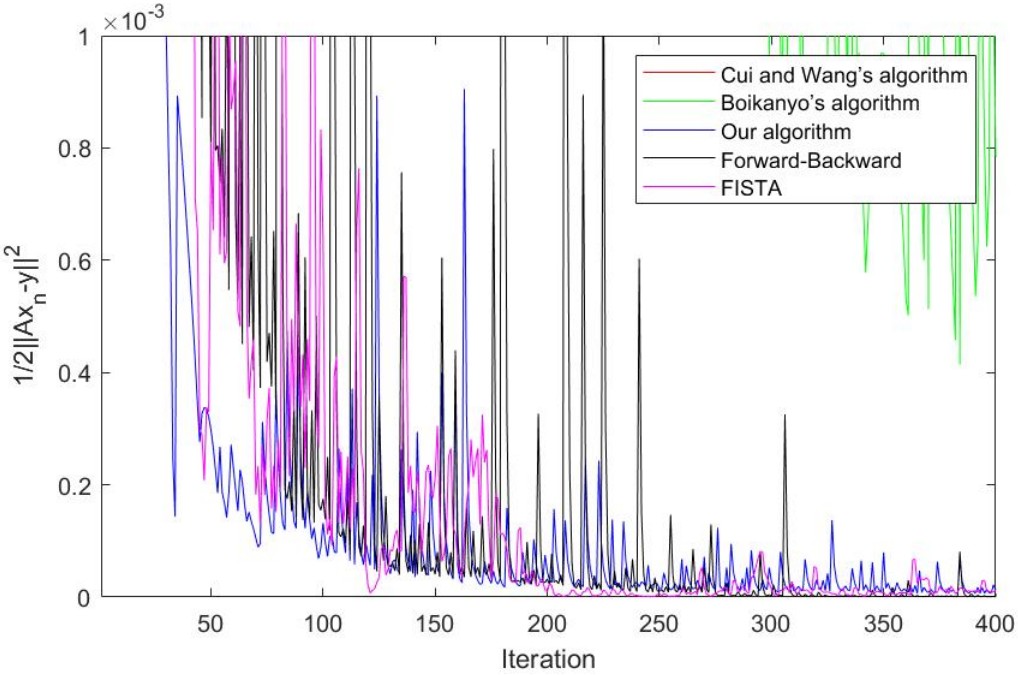

**Figure 3.** Case $N = t = 10$ and $k = 10$.

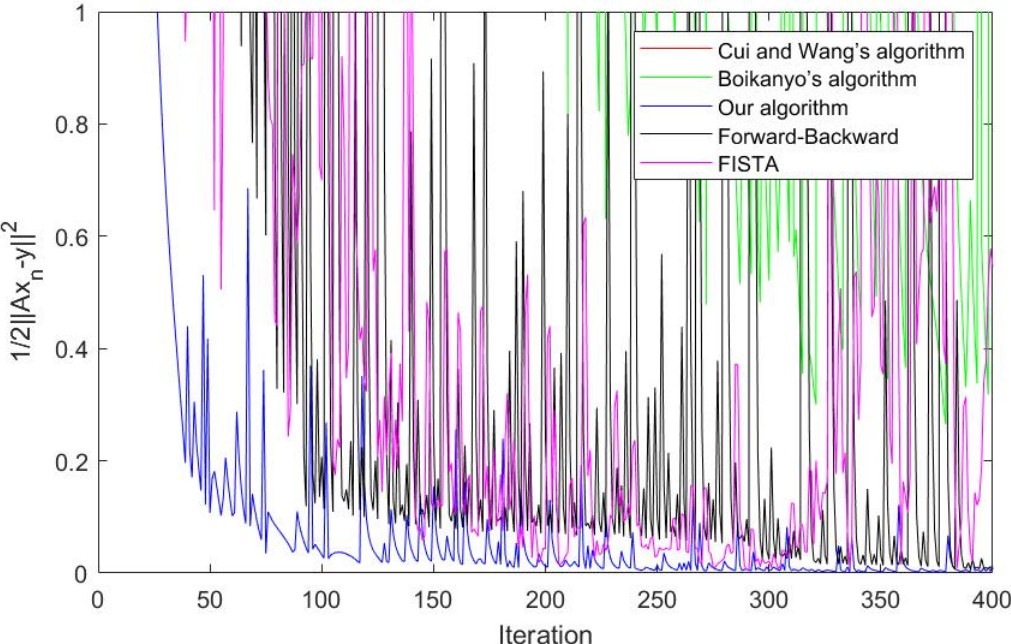

**Figure 4.** Case $N = t = 100$ and $k = 90$.

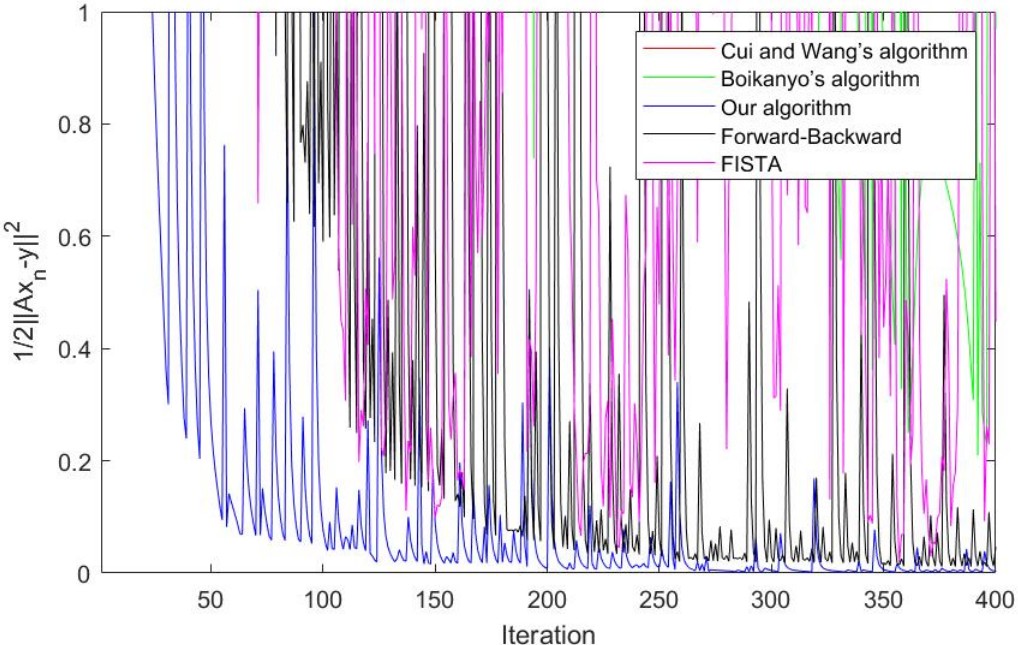

**Figure 5.** Case $N = t = 100$ and $k = 100$.

## 6. Conclusions

First, we proposed a new algorithm for demicontractive operators and improved that the sequence generated by our algorithm strongly converges to a solution of the problem (6). Moreover, our algorithm does not compute the norm of the bounded linear operator. Next, we obtained some results for many cases of operators such as a directed operator, a quasi-nonexpansive operator, a nonexpansive operator and a contraction operator.

**Author Contributions:** All four authors contributed equally to work. All authors read and approved the final manuscript. P.K. conceived and designed the experiments. W.J. performed the experiments. W.J. and Y.J.C. analyzed the data. K.S. and W.J. wrote the paper

**Funding:** Petchra Pra Jom Klao Ph.D. Research Scholarship (Grant No. 10/2560), TRF Research Scholar Award (Grant No. RSA6080047) and King Mongkut's University of Technology North Bangkok (Grant No. KKMUTNB-62-KNOW-40).

**Acknowledgments:** The first author should like to thank the Petchra Pra Jom Klao Ph.D. Research Scholarship and the King Mongkut's University of Technology Thonburi (KMUTT) for financial support. The authors acknowledge the financial support provided by King Mongkut's University of Technology Thonburi through the "KMUTT 55th Anniversary Commemorative Fund". This project Poom Kumam was partially supported by the Thailand Research Fund (TRF) and the King Mongkut's University of Technology Thonburi (KMUTT) under the TRF Research Scholar Award (Grant No. RSA6080047). Moreover, this research was funded by the King Mongkut's University of Technology North Bangkok, Contract no. KKMUTNB-62-KNOW-40.

**Conflicts of Interest:** The authors declare no conflict of interest.

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
