# Peer review of "A General Algorithm for the Split Common Fixed Point Problem with Its Applications to Signal Processing"

_mathematics, doi:10.3390/math7030226_

Round 1

Reviewer 1 Report

The authors in this work are studying the split common fixed point problem, an inverse problem that consists in finding an element in a fixed point set such that its image of linear transformation belongs to another fixed point set. The main results in this paper extend and improve some recent results. The main application of the results is an inverse problem in signal processing, showing the effectiveness of the proposed algorithm. In this paper 3 algorithms are presentd and tested, comparing their results in the last section. The author's Algorithm is no 3. The theoretical analysis of the elgorithm is analytical and detailed (Theorem 3.1).

Comments:

1) In my opinion the authors should also include comments comparing the 3 algorithms such as the computational complexity of each one, the rate of convergence and the number of iterations needed to reach a certain accuracy.

2) After figures 2-5, a more detailed explanation of the presented results should be stated.

Overall, the work is new and interesting,  If the authors do these minor changes it can be published.

Author Response

We revised all the problems suggested by Reviewer 1.

(1)  P 19, Section Conclusion: We add sentence that ”The number of iteration

of our algorithm is the least by the result’s experiment.”.

(2)  P 18, Example 5.3: We add sentence that ”This example show that number of iteration of our algorithm is the least under this parameters.”.

Reviewer 2 Report

1) page 15, after line 211: demicontrctive should be demicontractive.

2) recommendation for the References list:

Qamrul Hasan Ansari, Aisha Rehan and Jen-Chih Yao: Split feasibility and fixed point problems for asymptotically k-strict pseudo-contractive mappings in intermediate sense, Fixed Point Theory, Volume 18(2017), No. 1, 57-68.

Author Response

We revised all the problems suggested by Reviewer 2. (1) P 15, We changed demicontrctive to demicontractive. (2) P 2, We add References’s Q. H. Ansari et al.’s.

Reviewer 3 Report

Please find the attachment!

Author Response

We revised all the problems suggested by Reviewer 3.

Main issues:

(1) Algorithm’s H. Huimin is special case’s our algorithm where βn = 0. Al- gorithm’s H. Huimin and our algorithm difference because they generated distinct sequence xn. However, they converge strongly to a solution of the split common fixed point problem such that it is not necessary same point.

? ?? ??100? 0.1Example,Lety= 1.5 7 ,ε= 0.5 1 ,A= 1 2 3,αn= n,

? ?
β = 1 ,λ = 1,f(x)= x2 1 0 +?2 1 0?andt=10suchthat

nn2n2 4
is transpose. We will consider xOur,100 = ?1.5024 1.4672 0.8540?

generated by our algorithm and xH,100 = ?1.5034 1.0701 1.1177?gen- erated by algorithm’s H. Huimin et al.’s. They generate distinct sequence converge strongly to a solution of problem 5.3.

(2) In page 17, Section Application to signal Processing. We revised and add References’s H. Cui and L. Ceng.

Minor issues:

(1)  In page 2-4, 15-18: We have revised.

(2)  In page 7, We determine the inequality max{∥xn z,

βnf(z)z∥ ≤ max{∥x0 z, 1 f(z)z∥}+f(z)z? βn because

1η
xn+1 z∥≤max{∥xn z, 1 f(z)z∥}+βnf(z)z.

n=0

Thank you very much for valuable important suggestions of Reviewer 1,Reviewer 2, Reviewer 3 and Editor

Sincerely yours,

Wachirapong Jirakitpuwapat Poom Kumam
Cho Yeol Je
Kanokwan Sitthithakerngkiet

1 f(z) z∥} +

1η

1η
(3) In page 4: We add References’s Huimin et al.’s.

We revised all the problems suggested by Editor. (1) We revised twelve consecutive word or above.

2

Round 2

Reviewer 1 Report

I think that this work is quite interesting, and further theoretical research on this subject can be done by the authors or by others working in this field, based on this work. Therefore, I recommend this article for publication.

Author Response

We revised all the problems suggested by Reviewer 1.

Comments and Suggestions for Authors: I think that this work is quite interesting, and further theoretical research on this subject can be done by the authors or by others working in this field, based on this work. Therefore, I recommend this article for publication.

Answer: Thank you very much for comments and suggestions.

Reviewer 2 Report

1) page 4, lines 42-43: "we will propose an 43 Algorithm (3) for demicontrative operators" should be "we will propose the Algorithm 3 for demicontrative operators";

2) see also: Q.A. Ansari, A. Rehan, J.C. Yao: Split feasibility and fixed point problems for asymptotically k-strict pseudocontractive mappings in the intermediate sense, Fixed Point Theory 18(2017), 57-68

Author Response

We revised all the problems suggested by Reviewer 2.

Comments and Suggestions for Authors:

(1)  page 4, lines 42-43: ”we will propose an 43 Algorithm (3) for demicontra- tive operators” should be ”we will propose the Algorithm 3 for demicon- trative operators”

(2)  see also: Q.A. Ansari, A. Rehan, J.C. Yao: Split feasibility and fixed point problems for asymptotically kstrict pseudocontractive mappings in the intermediate sense, Fixed Point Theory 18(2017), 57-68

Answer:

(1) We will change it to ”we will propose the following Algorithm 4 for demi- contrative operators”. It appears below Algorithm 3 in page 4. We changed ”Algorithm 3” to ”Algorithm 4” because We add algorithm of Huimin et al.

(2) We refer ”Split feasibility and fixed point problems for asymptotically k- strict pseudocontractive mappings in the intermediate sense”. It appears at line 25 in page 2. That is [4].

Reviewer 3 Report

As I already mentioned in the previous report, the propose method is basically the same as the one

in [14]. It is a slight variant of [14] by adding a convex combination. 

Since (beta_n) is summable sequence,  the behaviour of x_{n+1} and y_n are the same. 

Some comments are: 

Provide a clear connection to [14]  after (1.10).

Shows that the parameter beta_n improve the convergence profile of [14].

The numerical results in section 5 should be compared to the existing method such as Forward-Backward splitting, accelerated proximal gradient method (Fista).

Author Response

We revised all the problems suggested by Reviewer 3.

Comments and Suggestions for Authors: As I already mentioned in the previous report, the propose method is basically the same as the one in [14]. It is a slight variant of [14] by adding a convex combination. Since (βn) is summable sequence, the behaviour of xn+1 and yn are the same. Some comments are:

(1)  Provide a clear connection to [14] after (1.10).

(2)  Shows that the parameter βn improve the convergence profile of [14].

(3)  The numerical results in section 5 should be compared to the existing method such as Forward-Backward splitting, accelerated proximal gradient method (Fista).

Answer:

(1)  Line 39 in page 4, We add the sentence ”In fact, our algorithm was changed the algorithm of Huimin et al. and the point u in Boikanos algorithm to the viscosity term and linear convex combination. The algorithm of Huimin et al. is a special case of our algorithm when βn = 0 and {λn} is a constant sequence.”.

(2)  Line 139 in page 17, We add the sentence ”Figure 2-5 show that βn improve the convergence profile of [13, 14]”.

(3)  We compared with the Forward-Backward and FISTA in Example 5.2. We add the sentence ”Moreover, we compared our algorithm with the Forward-Backward Splitting Algorithm [24] and the Fast Iterative Shrinkage- Thresholding Algorithm (FISTA) [25]. Sometimes, our algorithm con- verges faster than other algorithm Figure 4 and 5, but, sometimes, our al- gorithm converges slower than other algorithm Figure 2 and 3 . It depends on control condition. This experiment is an example for the convergence of some algorithms.” in line 140 page 17.

Round 3

Reviewer 3 Report

It is ok now